# Coastal Wetland Mapping Using Ensemble Learning Algorithms: A Comparative Study of Bagging, Boosting and Stacking Techniques

**Li Wen** [1,*] and **Michael Hughes** [1,2]

1   Science Division, NSW Department of Planning, Industry and Environment, Sydney,
    Australia Locked Bag 5022, Parramatta, NSW 2124, Australia; Michael.Hughes@environment.nsw.gov.au
2   School of Earth Atmosphere and Life Sciences, University of Wollongong,
    North Wollongong, NSW 2500, Australia
*   Correspondence: li.wen@environment.nsw.gov.au

**Abstract:** Coastal wetlands are a critical component of the coastal landscape that are increasingly threatened by sea level rise and other human disturbance. Periodically mapping wetland distribution is crucial to coastal ecosystem management. Ensemble algorithms (EL), such as random forest (RF) and gradient boosting machine (GBM) algorithms, are now commonly applied in the field of remote sensing. However, the performance and potential of other EL methods, such as extreme gradient boosting (XGBoost) and bagged trees, are rarely compared and tested for coastal wetland mapping. In this study, we applied the three most widely used EL techniques (i.e., bagging, boosting and stacking) to map wetland distribution in a highly modified coastal catchment, the Manning River Estuary, Australia. Our results demonstrated the advantages of using ensemble classifiers to accurately map wetland types in a coastal landscape. Enhanced bagging decision trees, i.e., classifiers with additional methods to increasing ensemble diversity such as RF and weighted subspace random forest, had comparably high predictive power. For the stacking method evaluated in this study, our results are inconclusive, and further comprehensive quantitative study is encouraged. Our findings also suggested that the ensemble methods were less effective at discriminating minority classes in comparison with more common classes. Finally, the variable importance results indicated that hydro-geomorphic factors, such as tidal depth and distance to water edge, were among the most influential variables across the top classifiers. However, vegetation indices derived from longer time series of remote sensing data that arrest the full features of land phenology are likely to improve wetland type separation in coastal areas.

**Keywords:** coastal wetland; saltmarsh; mangrove; fractional cover; sentinel-2; machine learning

## 1. Introduction

Coastal wetlands are a critical component of the coastal landscape, representing unique and important habitat for many species ranging from marine megafauna [1] to waterbirds (particularly shorebirds, [2]) to terrestrial mammals [3]. They also provide several ecosystem services, including the removal of nutrients and other pollutants, stabilising the shoreline, and carbon sequestration [4–6]. Nevertheless, coastal wetlands are among the most threatened ecosystems [7] and are disappearing worldwide at an alarming rate [8] due to land-use activities (e.g., conversion to aquaculture and agriculture, urbanisation; [9]) and other processes (e.g., climate change, sea level rise (SLR), decreased sediment supply [10–12]).

The growing awareness of the rapid loss of coastal wetlands and the services we derive from them has resulted in extensive studies to quantify the problem, understand its underlying causes, and assess possible solutions [9]. Among those studies, mapping wetland vegetation is crucial to

coastal ecosystem management [13,14], because vegetation communities are directly associated with ecosystem functions and services [15] and are suitable for assessing the conservation status of a site [16]. Recent advances in sensor design and geospatial analysis techniques are making remote sensing systems practical and attractive to manage coastal landscapes through mapping and monitoring coastal wetland vegetation [17] as well as other applications [14,18].

　　　Located at the interface of land and marine ecosystems, coastal wetlands represent the most diverse wetland types on the Earth and come in many forms including freshwater, brackish or saline [8]. Moreover, they are highly dynamic systems responding to variations in tidal and freshwater inflows [10]. Their high diversity and high degree of spatial heterogeneity may present a challenge for effectively mapping wetland vegetation [19,20]. To effectively differentiate major plant communities, most previous research relied on aerial photography (e.g., drone imagery) or commercial high spatial resolution satellite imagery such as quick Bird and Worldview-2 [21–23]. The high costs, limited coverage and irregular revisits associated with high spatial resolution imagery restrict their wider applications. Reliable tools to accurately discriminate coastal wetland types using the freely available medium-resolution satellite imagery (such as Landsat TM, ETM +, and OLI, or the more recent Sentinels-1 and 2), therefore, are of particular interest to researchers and resource managers involved in monitoring at the landscape or regional scale [24,25].

　　　In parallel with the advances in remote data acquisition sensors, for example, the launch of high-resolution satellites such as IKONOS (1 m), Quick Bird (0.61 m), and Worldview-2 (0.5 m), developing machine learning algorithms (MLA) for land cover classification (e.g., vegetation mapping) and modelling environmental data in general (e.g., Reference [26]) has become a major focus of the remote-sensing literature [25,27–29]. A wide variety of models have been developed in the field of machine learning since the term "Machine Learning" was coined in 1959 by Arthur Samuel [30]. Some of the popular MLAs including Random Forest (RF), Support Vector Machine (SVM), Neural networks (NN), and k-Nearest Neighbour (KNN), have been increasingly used for wetland classification and land cover/use mapping in general [21,31,32]; in many cases, they provide improved results [25] due to their capacity for handling more complex data better than classical statistical methods [33]. In the literature, the terms "classifier" or "learner" refers to any algorithm or model that produces a hypothesis about an object using a set of learning data. Here, we use these terms interchangeably as synonyms.

　　　Ensemble learning (EL) methods, which involve building and combining multiple learners, have been shown to produce better results and achieve improved generalisation compared with any of the constituent classifiers alone [34–36]. The ensemble approach is particularly useful when the amount of training data is small. With limited available training data, it becomes difficult to select an appropriate classifier. Ensemble algorithms could reduce the risk of choosing a poor classifier by averaging the votes of individual classifiers [37]. Many methods for constructing ensembles have been developed, but bagging, boosting and stacking are the commonly used techniques [38]. Briefly, bagging (also known as bootstrap aggregation; [39]) is the way to improve the stability and accuracy of MLA through training the same algorithm many times by using different subsets sampled from the training data (i.e., random samples with replacement, bootstrapping). RF, a modification of bagged decision tree (DT) that builds a large collection of independent trees by randomly choosing a subset of features (predictor variables) for each bootstrapped sample, can further improve predictive power [40]. The boosting method is an iterative process in which the first algorithm is trained on the entire training dataset and the subsequent algorithms are built by fitting the residuals of the previous algorithm, thus giving higher weight to those observations that were poorly predicted by the predecessor [41]. This mostly helps to reduce bias in the data set and to some extent leads to a reduction in variance as well [39]. Unlike bagging and boosting, which combine the results of similar base learners to build a strong learner through majority voting, stacking combines multiple layers of different MLAs using another MLA [42,43].

　　　Ensemble algorithms, such as RF [44] and Gradient Boosting Machine (GBM) [45], are now commonly applied in the field of remote sensing, and generally produce satisfactory results. The performance and

potential of other EL methods, such as extreme gradient boosting (XGBoost) [46] and bagged trees [47], are rarely compared and tested for wetland mapping. In this study, we test EL techniques such as bagging [48], boosting and RF using a wetland type map from a highly modified coastal landscape. To evaluate the performance of various EL methods, some widely used benchmark classifiers such as elastic net [49], decision tree (CART, [50]), Naïve Bayes classifiers (probabilistic machine learning models) [51], regularised discriminant analysis (RDA), SVM, NN and kNN are also tested and compared. Results from this investigation are intended to provide reliable information for the growing number of practitioners and resource managers engaged in regional and national coastal wetland monitoring and inventory projects.

## 2. Methods

### 2.1. Study Site

The Manning River Estuary (152.686° E and 31.879° S, Figure 1) is situated on the Mid North Coast of New South Wales, Australia. The estuary has a warm temperate climate with mild winter and summer. The mean-annual (1998–2018) rainfall is 1108 mm, mean maximum summer temperature is 28.5 °C, and mean minimum winter temperature is 7.5 °C [52]. It is a wave-dominated barrier estuary with an open, trained entrance (i.e., breakwater and bank structures for stabilisation). The Manning River and its tributaries are largely unregulated, with no major storages and weirs. However, the tidal regime has been substantially altered through dredging and entrance training walls [53], and the natural land cover has been extensively modified by land clearing for pasture and oyster aquaculture [54].

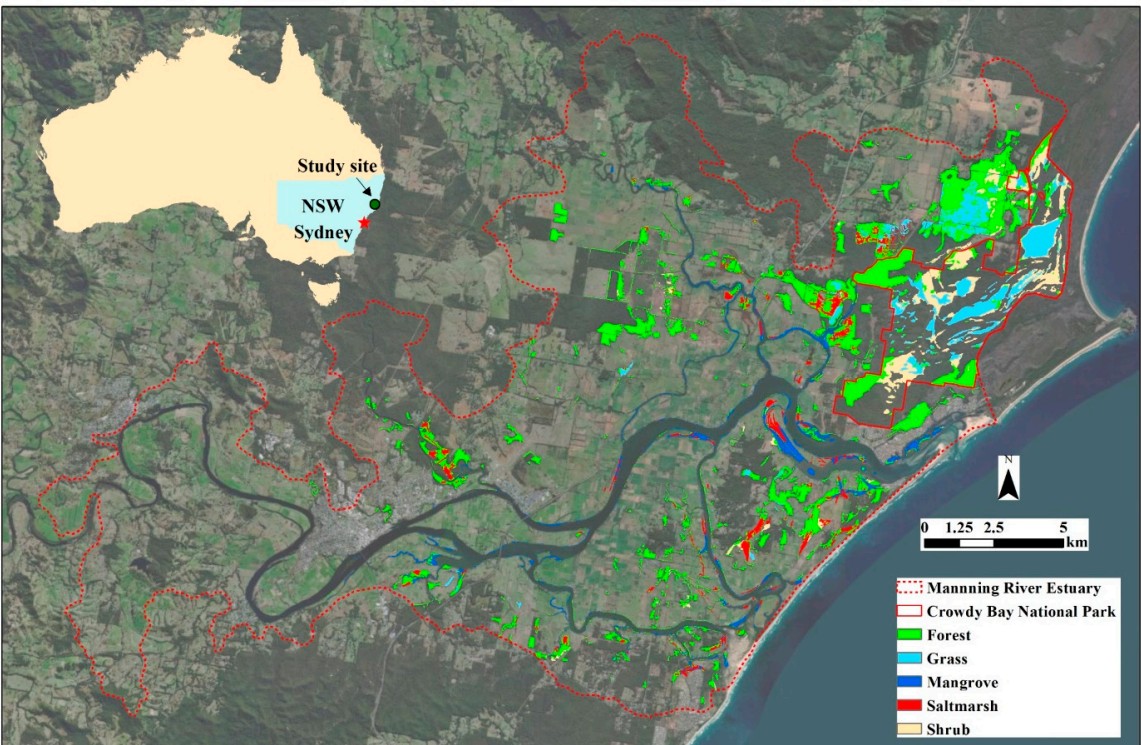

**Figure 1.** Manning River Estuary and the distribution of major wetland types. The Crowdy Bay National Park (>11,700 ha) is to the north of the estuary entrance. The broad vegetation classes are derived from aggregating vegetation classes recently mapped by EcoLogical in 2019 [55]. Background (aerial photo) shows extensive agricultural development. Inset map shows the location of the study site (marked) in New South Wales, Australia.

The Manning River Estuary has diverse relic wetlands including open freshwater lagoons dominated by floating macrophytes, large sedgelands, shrublands, forested wetlands dominated by tree species such as Swamp Oak (*Casuarina glauca*), Swamp Mahogany (*Eucalyptus robusta*), and Broad-leaved Paperbark (*Melaleuca quinquenervia*), saltmarshes, and open fringing mangrove forests and mudflats [55]. These wetlands are typically patchy and small, scattered across the foreshore, inter-tidal, riparian and floodplain areas; except the Crowdy Bay National Park (~11,700 ha) where extensive wet and dry heathlands occur. They provide essential ecological services (e.g., breeding and foraging habitats for waterbirds, fish and crustaceans, [54]) for the maintenance of local and regional biodiversity. Wetland conservation is a major challenge facing the management of the Manning River Estuary due to the increasing threats of SLR and land clearing [54]. Although seagrass beds are also important wetlands with conservation and economic values, they are excluded from this study because mapping seagrass requires different data sources.

*2.2. Data Source and Preparation*

2.2.1. Wetland Map

Using the recently available detailed wetland map [55], we created a wetland dataset of 9704 random points using stratified sampling [56]. We only sampled the wetland polygons with high credibility (i.e., verified by field survey or drone imagery, [55]) to ensure the data quality. The original map discriminates wetlands into 13 major groups based on dominant plant species. Since large-scale wetland classification with satellite imagery has practical limitations to the number of classes [45], we aggregated the 13 groups into five broad wetland types based on the vegetation structure and the salinity; including three freshwater types (i.e., forested wetland, shrubland wetland, and grassy wetland) and two saline wetlands (i.e., mangrove and saltmarsh). In addition, we randomly sampled 2888 points from other land covers using the land use/land cover map of NSW (NSW Landuse 2017, downloaded on 12 August 2019 from https://data.nsw.gov.au/data/dataset/nsw-landuse-2017) as "background" points. When creating background points, we excluded built-up areas, artificial waters, and intensive agriculture land (mainly modified pastures) because the hydrological connectivity in these areas is greatly interrupted. Most of the background points were sampled from terrestrial forests and bushlands. The final wetland dataset contained 12,592 data points with six classes.

2.2.2. Sentinel-2 Fractional Cover

Land Fractional Cover (FC) provides information about the proportions of green photosynthetic vegetation (PV), non-photosynthetic vegetation (NPV) and bare soil (BS) of the ground surface. The FC products are developed by the Joint Remote Sensing Research Program using Sentinel-2 imagery by the Copernicus Programme (https://www.esa.int/Our_Activities/Observing_the_Earth/Copernicus), and the algorithm is described in Reference [57]. In this study, we used only two of the three components of FC, i.e., PV and BS, since theoretically the three parts sum up to 100%. From https://data-access.jrsrp.org.au/, we obtained all available FC images with less than 5% cloud cover of the study area for the period of January 2019–December 2019 inclusively (a total of 36 images).

We first aggregated the 36 FC images to regular monthly time series of PV and BS using the maximum function. With the regular monthly time series, we calculated three variables: the maximum (max), minimum (min), and coefficient of variation (cov) values. Additionally, we calculated two additional variables to represent the trend of FC by fitting a quadratic regression for each grid:

$$\text{FC} = a + bT + cT^2 + \varepsilon \tag{1}$$

where $T = 1, 2, \ldots, 12$ is the month, and $b$ and $c$ are the linear and quadric trend, respectively.

A total of 10 FC metrics were computed for further analysis. The "*Raster*" package [58] within R version 3.6.1 [59] was used for all raster manipulations and calculations.

### 2.2.3. Hydro-Geomorphological Variables

We downloaded the 5 m LiDAR DEM (digital elevation model) covering the study site from Geoscience Australia (https://elevation.fsdf.org.au/). The 5 m DEM was corrected for streams and water bodies using field survey transects and has a fundamental vertical accuracy of at least 0.30 m (95% confidence) and horizontal accuracy of at least 0.80 m (95% confidence). The 5 m DEM was resampled to 10 m using bilinear interpolation. Several morphological variables were derived from the corrected DEM, including.

1 and 2 degree detrended DEM: the residuals of the linear regression of polynomial $(x, y)$ with a degree of 1 and 2 (Figure 2),

$$\mathrm{DEM}_{detrend1} = \mathrm{DEM}_{mean} + X + Y + \varepsilon \tag{2}$$

$$\mathrm{DEM}_{detrend2} = \mathrm{DEM}_{mean} + a_0 + a_1 X^2 + a_2 XY + a_3 Y^2 + a_4 X + a_5 Y + \varepsilon \tag{3}$$

where Equation (2) is the first-order detrended DEM and Equation (3) is the second-order detrended DEM, and $x$ and $y$ are the longitude and latitude of the centre point of a grid cell.

*CTI* (Compound Topographic Index)—a steady-state wetness index calculated using

$$cti = \ln\left[\frac{\alpha}{\tan\theta}\right] \tag{4}$$

where $\alpha$ = catchment area [(flow accumulation + 1) × (pixel area in m$^2$)], and $\theta$ is the ground slope angle in radians.

Local deviation from global (*LDFG*):

$$LDFG_i = \overline{y} - y_i \tag{5}$$

where $\overline{y}$ is the mean evaluation of the 3 by 3 window, and $y_i$ is the elevation of the focus grid.

Dissection: dissection in a continuous raster surface within a $3 \times 3$ window calculated using

$$dissection = \frac{h - h_{min}}{h_{max} - h_{min}} \tag{6}$$

where $h$ is the elevation of the focus grid and $h_{max}$ and $h_{min}$ are the maximum and minimum elevation within a $3 \times 3$ window.

*TPI* (Topographic Position Index): the difference between the value of a cell and the mean value of its 8 surrounding cells

$$TPI = h - h_{mean}. \tag{7}$$

These variables were computed using the 'spatialEco' package by Reference [60] and package 'Spatstat' by Reference [61].

In addition, with the modelled tidal plane of High High-Water Solstice Springs (HHWSS) acquired by Reference [62], we calculated two geohydrological variables as predictors: Distance to water edge (Distance) and water depth at highest tide (Tidal). To calculate the distance to water edge, we first defined the area of inundated land at HHWSS as all positive values of the difference between the tidal plane and the DEM. The inundated polygons were then converted to polylines, and a raster with 10 m resolution of Euclidean distance to polyline was created using ArcMap. The water depth at HHWSS was the magnitude of the difference between the tidal plane and DEM; therefore, negative values indicate higher grounds that are not inundated at HHWSS and positive values suggest lowlands that are flooded by tidal inundation.

To limit the effect of collinearity on model performance, we excluded the highly correlated variables. We calculated the Pearson correlation coefficients $R$ between every pair of the variables. For the pair with absolute $R$ greater than 0.70, we kept only one variable for further analysis. This procedure

resulted in a set of 17 variables; namely *CTI*, De_trend$_1$, De_trend$_2$, Tidal, Distance, *TPI*, *LDFG*, bs.trend.1, bs.trend.2, pv.trend.1, pv.trend.2, bs.max, bs.min, bs.cov, pv.max, pv.min, and pv.cov.

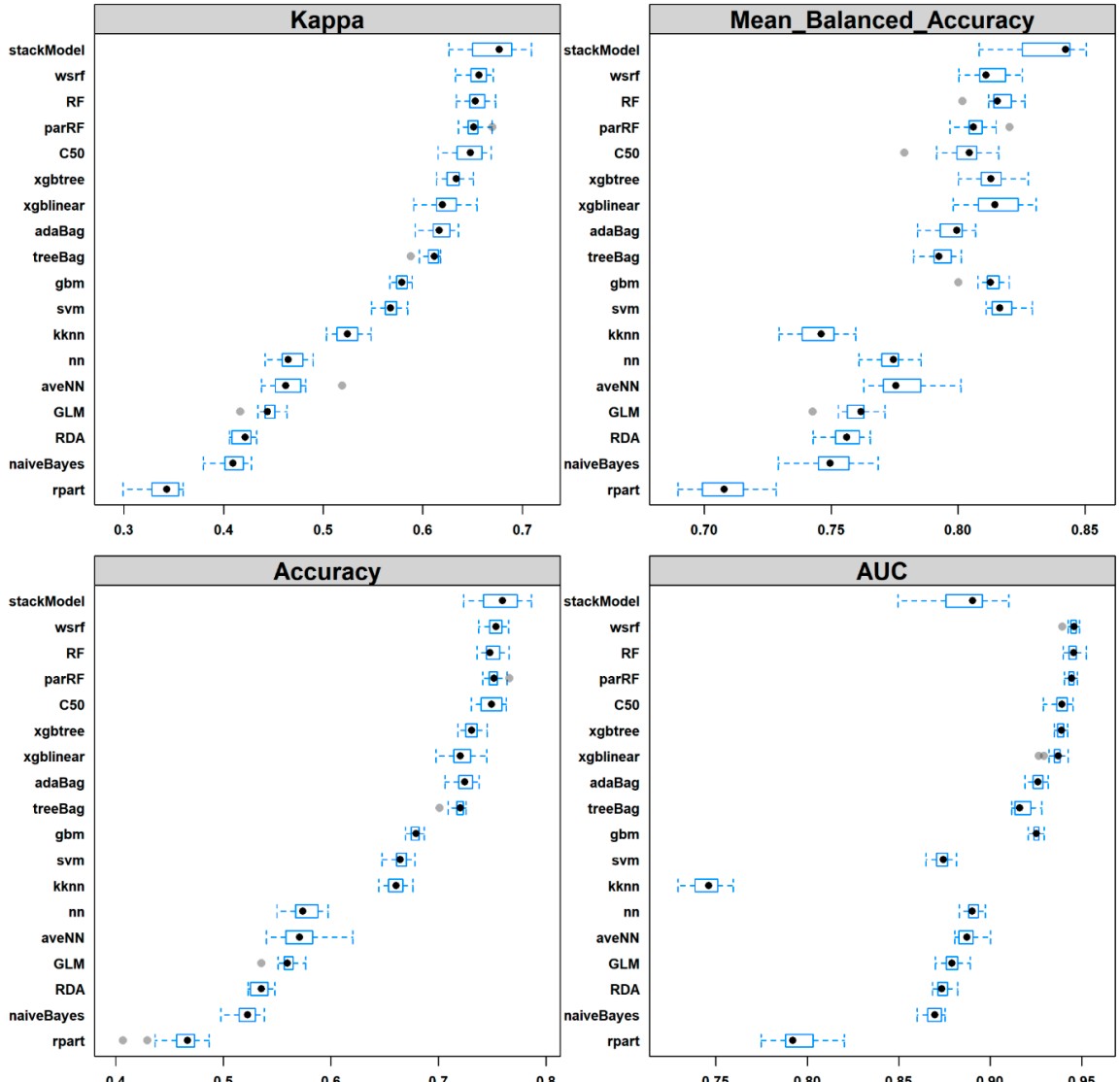

**Figure 2.** Training performance of the 18 machine learning models for wetland classification. Black dots are the mean value; box shows the 1st and 3rd quartiles; the dashed whiskers are the maximum and minimum values; grey dots are outliers based on the repeated-CV resampling. If there is no overlap of the quantile box, the difference is statically significant. See Table 1 for classifier description.

## 3. Modelling

We applied 18 classifiers (Table 1) to discriminate the wetland types in the Manning River Estuary. Details of the algorithms can be found in the listed references. Among these classifiers, seven are single and the rest are an ensemble. Each of the classifiers in Table 1 was tuned to find the best combination of parameters for optimising the prediction power. The entire dataset was split into training (75%, 9445 samples for model building) and testing (25%, 3147 samples for model verification) subsets using stratified random sampling. With the training dataset, we used the repeated K-fold cross-validation resampling method (repeat = 3 and k = 5 in this study) to tune the model parameters. In addition, since the classes are imbalanced, we adopted the "up-sampling" strategy, i.e., randomly sampled (with replacement) the minority class to be the same size as the majority class, to prevent biased prediction [63].

Final model selection was based on AUC (area under the receiver operating characteristic curve, see below). We used the caret package [63] for model training.

Our focus was to investigate the effectiveness of the three most popular EL methods (i.e., bagging, boosting, and stacking) to improve model performance and prediction accuracy, so we provide a brief description below.

### 3.1. Bagging

Bagging or bootstrap aggregating, introduced by Reference [39], is an ensemble method that involves training the same algorithm many times by using different subsets sampled from the training data. The final output prediction is then averaged across the predictions of all the sub-models. Bagging generally improves classification accuracy by decreasing the variance of the classification errors [39,64]. Breiman [39] suggests that bagging can improve accuracy greatly if perturbing the learning set can cause a significant change in the predictor constructed. Prediction of a test sample is made by taking the majority vote of the ensemble [65]. As each ensemble member is trained using a different set of samples, they differ from each other. By voting the predictions of each classifier, bagging seeks to reduce the error level due to the variance of the base classifier [39].

**Table 1.** The 18 classifiers evaluated in this study.

| Family/Classifier | R Package | Reference |
|---|---|---|
| Single | | |
| rpart: regression and classification tree | *rpart* | [67] |
| RDA: regularised discriminant analysis | *MASS* | [68] |
| kknn: weighted k-Nearest Neighbours | *kknn* | [69] |
| nn: feed-forward neural network | *nnet* | [70] |
| naiveBayes: the Bayes rule | *klaR* | [71] |
| svm: Support Vector Machines with Radial Basis Function Kernel | *kernlab* | [72] |
| GLM: elastic net regression | *glmnet* | [73] |
| Boosting | | |
| gbm: Stochastic Gradient Boosting | *gbm* | [74] |
| C50: rule-based models | *C5.0* | [75] |
| adaBag: Boosted CART | *adabag* | [76] |
| extreme gradient boosting: regression (xgbLinear) | *xgboost* | [46] |
| extreme gradient boosting tree (xgbtree) | *xgboost* | [46] |
| Bagging | | |
| RF: Random Forests * | *randomForest* | [39] |
| wsrf: weighted subspace random forest * | *wsrf* | [77] |
| parRF: Parallel Random Forest * | *e1071* | [78] |
| treeBag: Bagged CART | *Ipred* | [66] |
| avNNet: averaged neural networks | *NNet* | [69] |
| stackModel: stacking of top four classifiers with GBM | Various | [79] |

\* Extension of bagged decision trees with increased diversity through randomly selecting of subsets of predictor variables.

### 3.2. Boosting

In boosting, multiple models are trained sequentially, and each model learns from the errors of its predecessors [66]. The learning is achieved using adaptive resampling, in that a misclassified data point produced by an earlier classifier is selected more often than a correctly classified one. For each cycle in the training iteration, a weight is assigned to each training data point. In the next iteration, the classifier is obliged to concentrate on reweighted data points that were misclassified in the previous iteration. The final classifier is a weighted sum of the ensemble predictions.

### 3.3. Stacking

Stacking is concerned with combining heterogeneous machine learning models (base learners) using another data mining technique [80]. First, the base learners are trained, then a combiner (also called meta-classifier), is trained to make a final prediction based on the predictions of the base learner [81]. Such stacked ensembles tend to outperform any of the individual base learners (e.g., a single RF or GBM) and have been shown to represent an asymptotically optimal system for learning [42,78].

### 3.4. Performance Metrics for Classification Assessment

Many performance metrics or measures, such as Log-Loss, kappa coefficient, producer's and user's accuracy, and AUC, are given in the literature with the aim of providing a better measure of mapping accuracy [82]. Among these metrics, overall and individual class accuracies derived from the error matrix are the simplest, therefore have been widely used to evaluate model performance (e.g., References [45,83]). The kappa coefficient, which measures the difference between the actual agreement in a confusion matrix and the chance agreement, provides a better measure for the accuracy of a classifier than the overall accuracy as it takes into account the whole confusion matrix rather than the diagonal elements alone [84]. AUC is another widely used evaluation metric for checking the classification model's performance [85]. ROC (receiver operating characteristic curve) is a probability curve; AUC represents degree or measure of separability. It tells how much the model is capable of distinguishing between classes. The higher the AUC, the better the model is at predicting. In this study, we used the kappa coefficient, overall accuracy (OAA), mean balanced accuracy (MBA), and AUC to compare the models. In addition, for model validation, we also reported the class-level producer's and user's accuracy.

## 4. Results

### 4.1. Training Performance

Performance summary of the 18 tested classifiers is shown in Figure 2. In general, the rank of performance of the classifiers is consistent between the performance metrics kappa and OAA but not AUC and MBA (Figure 2). The ensemble classifiers had higher performance than the benchmark classifiers except for bagged neural networks (avNNet), which were among the less accurate performers (Figure. 2). The single tree classifier (rpart) had the worst performance in terms of the kappa coefficient of agreement (0.34), OAA (0.46), MBA (0.47). In addition, the resampling statistics indicated that rpart performed significantly worse than all other classifiers indicated by all assessment metrics except AUC (Figure 2). Naive Bayes was also among the low performers with a kappa of 0.41, and OAA and MBA of 0.52. Using the performance lexicon of Landis and Koch [86] for kappa coefficient, rpart was fair for discriminating wetland types whereas all other individual classifiers achieved moderate performance. All ensemble classifiers except gbm and avNNet had substantial power to separate wetland types [86].

Bagging classifiers with extended methods to add diversity between its ensemble members (i.e., wsrf, RF and parRF in this study) had significantly higher predictive power than boosting algorithms like xgbtree, xgblinear and adaBag, in terms of kappa, OAA and AUC (Figure 2). In addition, the differences between wsrf, RF and parRF were insignificant (Figure 2). In contrast, bagging algorithms based purely on bootstrapping (e.g., treeBag and avNNet) generally performed poorer than boosting ELs (such as xgbtree and adaBag) although the difference might not be significant (Figure 2). The rule-based ensemble trees (i.e., C50) were also among the highest performers, having comparable predictive powers with RF, parRF and wsrf.

Based on OAA, kappa and MBA, the stacking ensemble ranked as the best model although the difference was not significant for OAA and Kappa. The stacking ensemble was built with the four top classifiers (i.e., wsrf, parRF, rf, and C50) using GBM as the top layer learner. The predictions of the top four classifiers had a very low correlation (<0.38), thus the application of stacking is likely to have better performance [37].

*4.2. Testing Performance*

We validated trained classifiers with an independent testing dataset, and the summary of testing performance was presented in Table 2. As with the training performance, all ensemble classifiers showed good testing performance indicated by a high kappa coefficient of agreement (Kappa > 0.6, Table 2) between prediction and observation. Similarly, the individual classifiers all had moderate predictive power except rpart, which had fair predictive power (kappa = 0.33, Table 2).

**Table 2.** Summary of the testing performance of the 18 machine learning classifiers.

| Classifier | Accuracy | Kappa | MBA | Balanced accuracy (BA) | | | | | |
|---|---|---|---|---|---|---|---|---|---|
| | | | | BG | Forest | Grass | Shrub | Saltmarsh | Mangrove |
| stackModel | 0.7655 | 0.6817 | 0.8439 | 0.8817 | 0.8315 | 0.8699 | 0.8174 | 0.7701 | 0.8928 |
| RF | 0.7734 | 0.6865 | 0.8321 | 0.8826 | 0.8386 | 0.8606 | 0.8145 | 0.7198 | 0.8765 |
| wsrf | 0.7722 | 0.6833 | 0.8293 | 0.8846 | 0.8336 | 0.8490 | 0.8046 | 0.7299 | 0.8743 |
| parRF | 0.7684 | 0.6765 | 0.8220 | 0.8778 | 0.8343 | 0.8556 | 0.7893 | 0.7167 | 0.8583 |
| C50 | 0.7639 | 0.6693 | 0.8173 | 0.8734 | 0.8285 | 0.8578 | 0.7905 | 0.7015 | 0.8521 |
| xgbtree | 0.7426 | 0.6511 | 0.8286 | 0.8668 | 0.8154 | 0.8704 | 0.8049 | 0.7502 | 0.8640 |
| adabag | 0.7410 | 0.6408 | 0.8099 | 0.8598 | 0.8130 | 0.8493 | 0.7914 | 0.7121 | 0.8339 |
| xgblinear | 0.7350 | 0.6437 | 0.8326 | 0.8668 | 0.8052 | 0.8591 | 0.8234 | 0.7741 | 0.8671 |
| treeBag | 0.7363 | 0.6347 | 0.8098 | 0.8562 | 0.8077 | 0.8446 | 0.7888 | 0.7305 | 0.8307 |
| gbm | 0.6969 | 0.6035 | 0.8286 | 0.8434 | 0.7800 | 0.8577 | 0.8315 | 0.7839 | 0.8751 |
| svm | 0.6740 | 0.5797 | 0.8229 | 0.8485 | 0.7632 | 0.8429 | 0.8147 | 0.8114 | 0.8564 |
| kknn | 0.6657 | 0.5346 | 0.7523 | 0.8196 | 0.7618 | 0.8092 | 0.7288 | 0.6626 | 0.7317 |
| rda avNNet | 0.6060 | 0.5002 | 0.7869 | 0.7949 | 0.7236 | 0.7947 | 0.7825 | 0.7684 | 0.8575 |
| nn | 0.5875 | 0.4814 | 0.7828 | 0.8088 | 0.6913 | 0.8280 | 0.7816 | 0.7552 | 0.8320 |
| GLM | 0.5647 | 0.4500 | 0.7625 | 0.7431 | 0.7057 | 0.7799 | 0.7705 | 0.7393 | 0.8367 |
| rda | 0.5405 | 0.4272 | 0.7620 | 0.6870 | 0.7085 | 0.7921 | 0.8090 | 0.7683 | 0.8073 |
| naiveBayes | 0.5297 | 0.4201 | 0.7611 | 0.7001 | 0.6912 | 0.7901 | 0.8147 | 0.7340 | 0.8364 |
| rpart | 0.4474 | 0.3301 | 0.7080 | 0.7090 | 0.6203 | 0.7449 | 0.7006 | 0.6878 | 0.7851 |

See Table 1 for model description. Balanced accuracy = (Sensitivity + Specificity)/2. MBA=mean balanced accuracy.

In terms of OAA and Kappa coefficient, RF was the best classifier followed by wsrf. However, the stacking EL ranked first in terms of MBA (Table 2). At the wetland type level, all ensemble classifiers achieved high discriminating power for background, forested wetlands, mangroves, and grassy wetlands (BA > 0.8, Table 2). In contrast, all those classifiers were less effective at separating saltmarsh, with the highest balanced accuracy of 0.7701 achieved by stacking ensemble. The discrimination power for freshwater shrubland varied with classifiers, and some individual classifiers such as SVM, Naïve Bayes and rda achieved satisfactory results (BA > 0.8).

SVM was superior to other tested methods in discriminating the minority classes (i.e., shrub wetlands and saltmarshes, which have less distribution than other wetland types in the Manning River area, Figure 3). One boosting classifier, GBM, also achieved good performance in discriminating minority classes but was less effective than other ensemble classifiers in classifying the majority classes (such as background and forested wetlands, Figure 3) (Table 2).

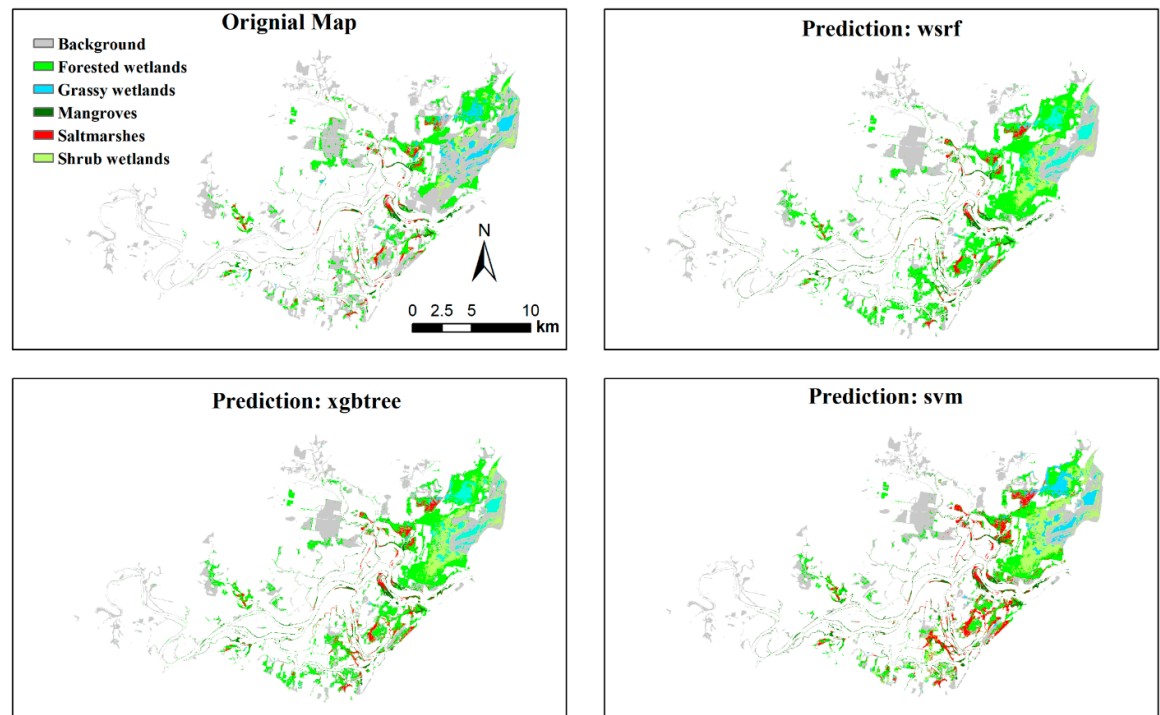

**Figure 3.** Comparison of mapped and predicted wetland distribution in Manning River Estuary. The original wetland map was an aggregation of the wetland map by Reference [55] Over-estimation of shrub wetlands and saltmarshes in the Crowdy Bay National Park at the upper right of the estuary is evident for all ensemble classifiers.

### 4.3. Predicted Wetland Type Maps

We selected three classifiers, (wsrf for bagging, xgbtree for boosting, and svm for the individual classifier, they were the best performers among their corresponding category) to predict the wetland type distribution in the study area. In comparison with the original map, all classifiers over-estimated wetland distribution at the northern part of the Crowdy Bay National Park and areas adjacent to the northern border of the park categorised as "grazing native vegetation" in the NSW land use map. In these areas, many parts of the background were misclassified as forested wetlands, and to less extent, as shrub wetlands, but the misclassification between wetland types was uncommon (Figure 3). In other areas, where agricultural land uses dominate, and wetlands occur as relic small patches or linear features along the riverbanks and the edge of modified pastures, the predictions were relatively accurate in terms of the occurrences and types of wetland although overestimating the size was common (Figure 3).

The single classifier svm tended to misclassify background to saltmarsh, and to less extent, mangrove, across the estuary. The overestimation was more profound in Mitchells Island near the entrance (Figure 3). In addition, the overestimation mainly occurred at the small openings within a large forest (or shrub) patch.

The locations where the three algorithms agree and disagree are shown in Figure 4. In most of the locations (70% of the grids), all three classifiers agree in class allocation. There are also many locations (27%) where two classifiers agreed with each other, and a small proportion of grids (3%) where there was no agreement between the predictions of the classifiers. The disagreements appeared randomly distributed across the study area.

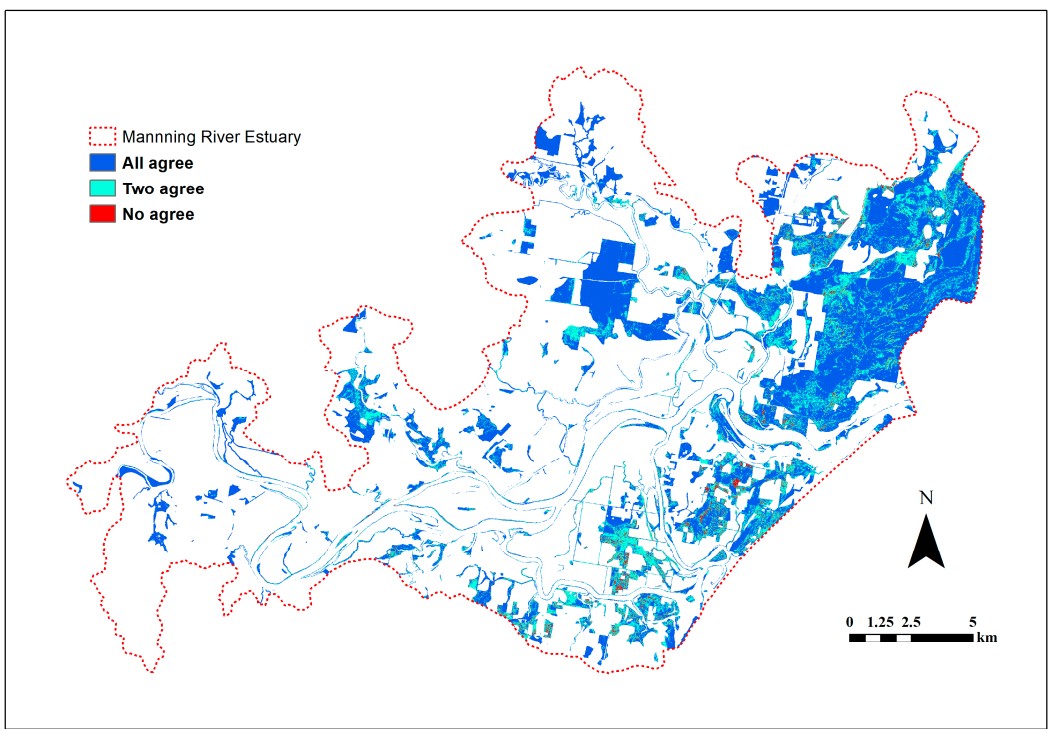

**Figure 4.** Map showing the locations where three classifiers (wsrf, xgbtree and svm) agree and disagree in predicting wetland types. The three classifiers predicted the same types for the majority of grids.

## 4.4. Variable Importance

We extracted the contribution of predictor variables to model performance for four classifiers: one boosting tree (xgbtree), one bagging tree (RF), and two individual classifiers (NN and rpart). With the exception of one tree classifier, the contribution of tidal height was far more important than other predictors (Figure 5). Other hydro-geomorphological variables, such as distance to the water's edge and LDFG, also made a large contribution to the discriminative power of ensemble classifiers. The contribution of predictors derived from Sentinel-2 fractional cover was also substantial, especially for the individual classifiers, which are the top predictor for rpart (Figure 5).

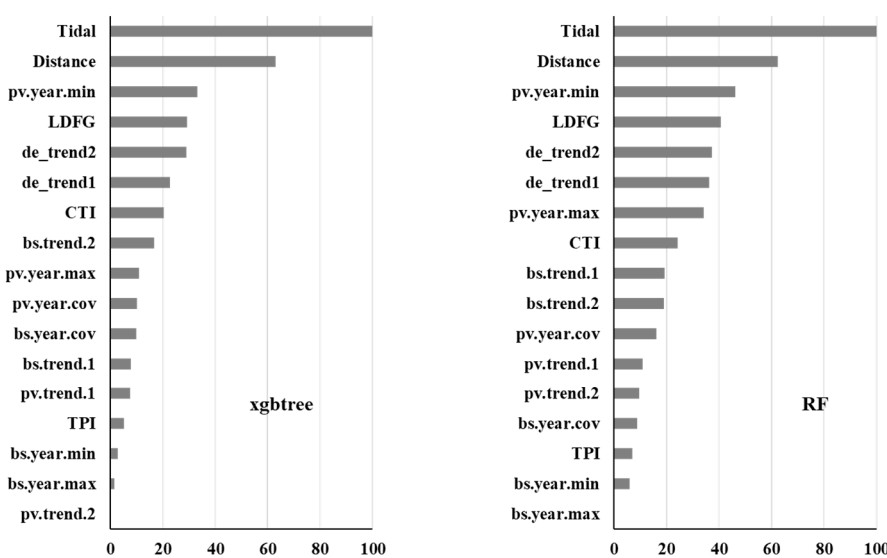

**Figure 5.** *Cont.*

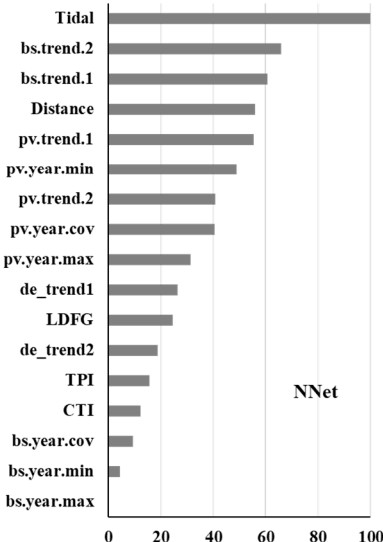
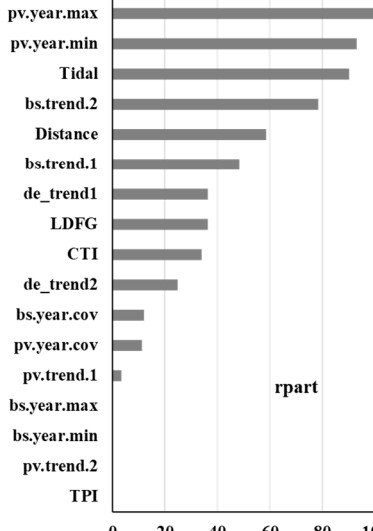

**Figure 5.** The importance of the predictor variables for selected classifiers. The importance value has been scaled to range between 0 and 100.

## 5. Discussion

The recent rapid progress in ML has resulted in a variety of algorithms, including individual classifiers (e.g., ANN, SVM and DT), boosting and bagging ensembles (such as adaboost, GBM and RF), and stacking ensembles. Some of these methods (e.g., RF and SVM) have been widely applied in remote sensing classification [28,29] while the applications of others (e.g., extreme gradient boosting) are relatively new with limited applications to date [64,87]. Therefore, comparative studies that evaluate the predictive power of classifiers are necessary to guide application-oriented research [87]. This is particularly true for mapping wetland types in dynamic environments such as coastal catchments, where the loss, transition and degradation of wetlands are accelerating [88]. In this study, we provided a comprehensive comparison of some advanced ensemble techniques in discriminating wetland types in a highly modified estuary using the combination of Sentinel-2 fractional cover and hydro-geomorphological variables. Our findings contributed to improving the applicability of ML for detailed coastal wetland mapping.

### 5.1. Overall Performance of Boosting, Bagging and Stacking Classifiers

There is a general consensus in the literature that ensemble classifiers outperform individual approaches in many applications [36,37,39,42,78]. Our results confirm this belief by indicating that all ensemble classifiers, except avNNet, had a significantly higher performance than the individual classifiers in both training and testing based on several performance metrics; OAA, kappa and MBA. Moreover, our results suggest that bagging classifiers with additional processes to add diversity between its ensemble members (for example, randomly selected a subset of predictor variables in RF) achieved significantly higher predictive accuracy compared to the other classifiers, including bagging methods based purely on random resampling (e.g., treeBag) and all boosting techniques. This finding suggests that ensemble diversity may be the key factor affecting overall classification performance [89]. For the high-performance ensemble classifiers, our results also show that differences in discrimination capacity were not significant (there were overlaps in the 1st and 3rd quartile between the performance metrics, Figure 2). Note that the rule-based bagging C50 produced comparable accuracy to other high performing classifiers.

Through combining different independent classifiers using a meta-classifier [81], stacking ensembles is one of the most active research areas in ML [29]. Many studies have demonstrated that stacking ensemble classifiers could improve the classification accuracy in comparison to their member classifiers [34,41,78,90].

In this study, we used GBM as the meta-learner to combine four top classifiers. Although the stacking model was among the top performers, the assessment using the four performance metrics (i.e., OAA, Kappa, MBA and AUC) was inconsistent for both training and testing (Figure 2 and Table 2). While the stacking model has significantly higher training MBA than any of its component classifiers, the AUC was significantly lower. In addition, the training performance of the stacking model is comparable (i.e., no significant difference) with the member classifiers' Kappa and OAA. Thus, our study did not demonstrate the superiority of stacking ensemble over well-tuned bagging or boosting classifiers. Nevertheless, we only tested one stacking model, and the potential for using other algorithms [91] and other deep learning methods (such as a deep belief network, [92]) to fuse the independent classifiers, and the effects of classifier selection [78,91], were not evaluated. A further comprehensive study is needed to evaluate fully the potential of stacking ensemble algorithms in mapping the distribution of wetlands in coastal landscapes.

### 5.2. Individual Class Performance

At the class level, the top ten classifiers (all ensembles) achieved high accuracy for background, forest, grassy wetland, and mangrove. However, all these classifiers were less effective at discriminating the less common classes (i.e., shrubland and saltmarsh), even though we adopted the "up-sampling" strategy [63] in model tuning to limit the bias towards the majority class [93]. For example, the producer's accuracy (results not shown) of the top 10 classifiers for saltmarsh was particularly low, ranging from 42% (C50) to 62% (gbm). Learning from imbalanced data sets is a challenging problem in knowledge discovery in many real-world applications [94] and remains an active research area in land use land cover classification [95] including wetland mapping.

Of the targeted classes, saltmarsh is the least abundant wetland type. It often occurs as a small patch, surrounded by forest or shrubland (Figure 3). The highest performance of MBA 81% (producer's accuracy 70%) was achieved by svm. However, visual comparison of svm prediction and the original map (Figure 3) revealed that svm overestimated saltmarsh distribution (mainly the extent of wetland) at many locations. The overestimation was mainly due to misclassification of the adjacent background (and to less extent shrubland) as saltmarsh, reflecting the importance of mapping scale and spectral resolution of the remotely sensed data [96].

### 5.3. Variable Importance

In this study, two groups of predictor variables were combined to model wetland distribution. By using the statistical summary calculated from the land fractional cover time series derived from multi-temporal Sentinel-2 data, this study followed earlier studies which linked the land phenology with plant community type (e.g., References [32,97]). Including hydro-geomorphic factors, which force typical zonation of vegetal settlement according to the plant tolerance of salinity and inundation [8,10], can enhance the discrimination capacity of the classifiers [98]. The importance of tidal depth, which was calculated as the difference between the DEM and HHWSS tidal plane, and distance to water edge, was consistent across most classifiers, confirming the key roles of inundation regime and salinity gradient in determining the coastal vegetation distribution [8].

The contribution of metrics summarising the time series of fractional cover to the discriminating power varied across the classifiers and was generally lower than that of hydro-geomorphological variables in ensemble models (but not in individual classifiers). This is surprising since many previous studies demonstrated the dominant contribution of vegetation indices derived from remote sensing data in vegetation mapping (e.g., References [41,98–100]). Due to data availability, the FC variables in this study were based on the Sentinel-2 images in 2019 (to match the time of wetland mapping), which was extremely dry (total rainfall 453 mm, less than half of the long-term mean, [52]). Under drought conditions, coastal vegetation, especially freshwater species, are stressed [101]. Therefore, plant growth characteristics such as growth rate, growth seasonality and peak growth might not be as distinguishable between vegetation types as they would otherwise be. Future research using longer time series of

FC data that arrest the full features of land phenology are likely to provide enhanced and improved wetland distribution maps in coastal areas.

## 6. Conclusions

This study has demonstrated the advantages of using ensemble classifiers to accurately map wetland types in a coastal landscape. All ensemble models except avNNet (averaged neural networks) had significantly higher performance than individual classifiers. Enhanced bagging DTs, i.e., classifiers with additional methods to increase ensemble diversity such as random forest and weighted subspace random forest, show comparably high predictive power. For the stacking method investigated in this study, i.e., using stochastic gradient boosting (GBM) to combine the four non-correlated top performers, our results are inconclusive, further comprehensive study is encouraged. Our findings also suggest that the ensemble methods are less effective to discriminate minority classes (saltmarsh in our case) in comparison with more common classes. Finally, the variable-importance results indicate that hydro-geomorphic factors, such as tidal depth and distance to water edge, are among the most influential variables across the top classifiers. However, vegetation indices derived from longer time series of fractional cover data that reflect the full features of land phenology are likely to improve wetland type separation in coastal areas.

**Author Contributions:** Conceptualisation, L.W. and M.H.; Methodology, L.W. and M.H.; Validation, L.W.; Formal analysis, L.W.; Writing—Original Draft Preparation, L.W.; Writing-Review and Editing, L.W. and M.H.; Funding Acquisition, M.H. All authors have read and agreed to the published version of the manuscript.

**Funding:** This research received no external funding.

**Acknowledgments:** We thank Karen Bettink from Mid Coast Council for directing us to council's vegetation mapping program, the vegetation map used in this study [55], and her ongoing interest in wetland mapping.

**Conflicts of Interest:** The authors declare no conflict of interest.

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
