# Peer review of "Coastal Wetland Mapping Using Ensemble Learning Algorithms: A Comparative Study of Bagging, Boosting and Stacking Techniques"

_remotesensing, doi:10.3390/rs12101683_

Round 1

Reviewer 1 Report

The manuscript presents a comparison of 18 machine learning classification algorithms, using an Australian estuary and 6 classes as a case study. The paper provides sufficient detail in the description of the types of learners (single, boosting, bagging, and stack). For predictor layers, the authors rely primarily on derived characteristics from a LiDAR DEM.  Curiously, only one derived satellite-based metric was included (timeseries fractional cover). Four metrics are used to compare model performance.  

The comparison of machine learning methods is robust and provides valuable information to the research community.  I have some concerns regarding the input data used to train the various models, however. The omission of spectral indices beyond FC is puzzling, as one would expect vegetation or soil metrics to be valuable in vegetation classification. The authors do acknowledge that a longer timeseries vegetation metrics may improve classification accuracy. Metrics that capture the photosynthetic activity of plants certainly seem to be appropriate here, since cover alone may not be a meaningful metric for phenology in all species. While I wouldn’t necessarily expect the relative performance of various classification algorithms to be dependent on the quality or type of input data, the lack of more commonly used predictor metrics may limit broad utility of comparisons.  

The authors use of a DEM to generate a wide range of predictor variables is notable and interesting.  However, there are issues with relying so heavily on a DEM. First is the issue of DEM accuracy. LiDAR-derived bare earth DEMs can have substantial vertical bias caused by interference of the LiDAR pulse by dense vegetation, an issue that becomes especially noticeable in tidal wetlands, which support dense vegetation and are sensitive to relatively small vertical errors given the influence of tides [what is the tidal range of estuary?].  Additionally, the amount of bias can vary substantially by dominant plant species or in this case, wetland class, which may affect the success of the classification. Was the lidar flown during leaf-off conditions and what was the point density?  Second, is the issue of the elevation in a discontinuous landscape. All classification methods over-predicted wetland extent, which suggests the elevation metrics were not able to distinguish lowlying areas from wetlands; was connectivity considered in the Tidal metric, or in any way?  

The main point of the paper is strong – the comparison of classification methods is robust and provides valuable information to other researchers.  I recommend a minor revision that addresses the above concerns regarding the predictor layers.

Author Response

We appreciate your supportive comments and constructive suggestions.

We agree with your comment on using just fractional cover metrics might have limited the performance of the classifiers. Besides the original reflectance data, there are numerous satellite-derived vegetation indices (VI), notably the Normalized Difference Vegetation Index (NDVI), Soil-Adjusted Vegetation Index (SAVI), leaf area index (LAI) and Enhanced Vegetation Index (EVI), are available; and many studies have demonstrated the value of VI for discriminating vegetation types. The fractional cover (FC), which describes a vertical projection of the areal proportion of a landscape occupied by green and brown vegetation and bare soil, is mainly used in monitoring vegetation growth status and crop yields, and in earth surface process simulations. It could be very useful to characterize the spatial pattern of vegetation types as well. In this study, we used metrics based on time series of two parts of the FC (i.e. green vegetation and bare soil, as 2019 is extremely dry and we expected different vegetation types may have distinguishable responses) to discriminate wetland types, and found they had considerable contribution to model predictive power, especially for individual classifiers. Please note that we revised the statement on the FC in the Discussion. As you rightly pointed out, the main aim of the study is to compare the performance of the ensemble classifiers. The results should be relatively robust regardless of input data layers.

For the issue of DEM accuracy, the nation-wide 5 m DEM from Geoscience Australia was resampled from 1 m LiDAR-derived DEM, which is quality assured and has a fundamental vertical accuracy of at least 0.30 m (95% confidence). The resampling could further smooth out the survey errors. In addition, the DEM for waters (rivers and lakes) was corrected with field survey transects by an independent consultant. The 0.3 m vertical error is much lower the mean tidal range of ~ 1 m and mean spring tidal range of ~ 1.4 m (maximum range less than 2 m), therefore may have limited impacts on the two geohydrological variables (i.e. tidal height and distance to water edge). However, the elevation error caused by vegetation (i.e. over estimation in dense vegetated areas such as forests) is indeed an issue. As you pointed out, this bias may be responsible for the over-prediction the extent of grassy wetlands. Unfortunately, without extensive field survey, there is no systemic way to correct the bias. We addressed these limited in the Discussion. In addition, we can’t provide such details of LiDAR survey as the cloud density and flight time as the information is not available from the metadata from Geoscience Australia.   

Reviewer 2 Report

This paper applies the existing ensemble algorithms to deal with the coastal wetland mapping problem. Also, some data collected from Australia is presented. Although the paper is valuable in practice, its academic novelty is pretty limited. I suggest the authors could pay more attention to their main contributions. For example, the manuscript could be rewritten from the published data aspect.

Author Response

Thank you very much for the positive comments.

We did have a practical intention when design the research. To emphasis the purpose of the study, we added the following sentence in the end of Introduction:

“Results from this investigation are intended to provide reliable information for the growing number of practitioners and resource managers engaged in future regional and national coastal wetland monitoring and inventory projects”.

Also, we checked the English (grammar, expression and the consistent use of terms) throughout the manuscript and made a number of changes.

Round 2

Reviewer 2 Report

All of the issues have neem modified. I suggest the current version can be accepted.